# Evidence-Aware Entropy Decomposition For Active Deep Learning

## Abstract

We present a novel multi-source uncertainty prediction approach that enables deep learning (DL) models to be actively trained with much less labeled data. By leveraging the second-order uncertainty representation provided by subjective logic (SL), we conduct evidence-based theoretical analysis and formally decompose the predicted entropy over multiple classes into two distinct sources of uncertainty: vacuity and dissonance, caused by lack of evidence and conflict of strong evidence, respectively. The evidence based entropy decomposition provides deeper insights on the nature of uncertainty, which can help effectively explore a large and high-dimensional unlabeled data space. We develop a novel loss function that augments DL based evidence prediction with uncertainty anchor sample identification through kernel density estimation (KDE). The accurately estimated multiple sources of uncertainty are systematically integrated and dynamically balanced using a data sampling function for label-efficient active deep learning (ADL). Experiments conducted over both synthetic and real data and comparison with competitive AL methods demonstrate the effectiveness of the proposed ADL model.

## 1 Introduction

Deep learning (DL) models establish dominating status among other types of supervised learning models by achieving the state-of-the-art performance in various application domains. However, such an advantage only emerges when a huge amount of labeled training data is available. This limitation slows down the pace of DL, especially when being applied to knowledge-rich domains, such as medicine, biology, and military operations, where large-scale labeled samples are too expensive to obtain from well-trained experts. Meanwhile, active learning (AL) has demonstrated great success by showing that for many supervised models, training samples are not equally important in terms of improving the model performance (Settles, 2009). As a result, a carefully selected smaller training set can achieve a model equally well or even better than a randomly selected large training set.

An interesting question arises, which is whether DL models can be actively trained using much less labeled data. Recent efforts show promising results in this direction through Bayesian modeling (Gal et al., 2017) and batch model sampling (Sener & Savarese, 2018). However, as DL models are commonly applied to high dimensional data such as images and videos, a fundamental challenge still remains, which is how to most effectively explore the exponentially growing sample space to select the most useful data samples for active model training. Existing AL models usually leverage the model provided information, such as estimated decision boundaries or predicted entropy for data sampling. However, the deep structure and the large number of parameters of DL models make model overfitting almost inevitable especially in the early stage of AL when only very limited training data is available. As a result, the model may provide misleading information that makes data sampling from a high-dimensional search space even more difficult. Besides a high dimensionality, complex data may contain a large number of classes and data samples from certain classes may be completely missing. Such situations are quite common for domains, such as scientific discovery (e.g., gene function prediction) and anomaly detection. AL models should be able to effectively discover these out of distribution (OOD) samples for labeling in order to achieve an overall good prediction performance.

Uncertainty sampling has been one of the most commonly used pool-based AL models. In particular, a model chooses the data sample that it is least certain about. Thus, once the sample is labeled, model uncertainty can be significantly reduced. As an information-theoretic measure, entropy provides a

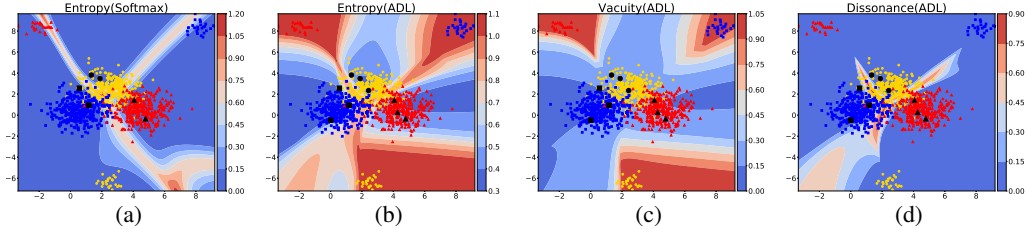

Figure 1: A dataset consists of three mixtures of Gaussian's (shown in red, blue, and yellow, respectively), each of which has a large and small clusters of data samples. (a) Softmax predicted entropy; (b) ADL predicted entropy; (c) ADL predicted vacuity; (d) ADL predicted dissonance.

general criterion for uncertainty sampling. Some commonly used sampling methods, including least confident and margin based strategies, are equivalent to entropy-based sampling in binary classification (Settles, 2009). It is also natural and straightforward to generalize to multi-class problems.

A key challenge of entropy-based sampling for AL is that the predicted entropy may be highly inaccurate, especially in the early state of the AL. Such an issue may become more severe when training a neural network (NN)/DL active learner due to model overfitting as described above. Figure 1(a) shows the predicted entropy by an NN active learner trained using nine labeled data samples, which are in black color and evenly distributed in three classes. The standard *softmax* layer is used in the output layer to generate class probabilities over three classes, each of which is a mixture of two Gaussian's. It turns out that all the data samples in the three small clusters located in the top left, top right, and bottom center, are wrongly predicted with high confidence, as indicated by the low entropy. As a result, data samples from these three clusters are less likely to be selected for labeling. In contrast, the data samples that are close to the center of the three major clusters are more likely to be selected. However, labeling these samples will have the effect of fine-tuning a wrongly predicted decision boundary, leading to a much higher (but less effective) labeling cost.

Figure 1(b) shows the the result from the proposed active deep learning (ADL) model. While the samples from the small clusters are still wrongly predicted due to lack of training data, they are predicted with a much lower low confidence as indicated by the high entropy. However, even with a more accurately predicted entropy, the active learner may still sample from the center of the three major clusters as it is still assigned a high entropy along with the areas that cover the three smaller clusters. By performing a fine-grained analysis of uncertainty under the subjective logic (SL) framework (Jøsang, 2016), we formally *decompose entropy into two distinct sources of uncertainty: vacuity and dissonance*, which are caused by lack of evidence and conflict of strong evidence, respectively. By putting the vacuity and dissonance as shown in Figures 1(c) and (d) together, it is interesting to see that we recover the entropy as shown in Figure 1(c), which empirically verifies our theoretical results. Entropy decomposition provides further insights on the sources on uncertainty, which is instrumental to guide the data sampling process. Intuitively, given the dataset in Figure 1, an effective sampling strategy will first choose samples from the three small clusters according to vacuity in the early stage of AL to properly establish the shape of the decision boundary. It can then fine-tune the decision boundary by sampling according to dissonance. Such an uncertainty-aware sampling strategy will be critical for a high-dimensional space with multiple competing classes where data samples are scarcely distributed and the decision boundary becomes more complicated.

Our major contribution is threefold: (1) decomposition of entropy through evidence-based theoretical analysis of belief vacuity and belief dissonance under the SL framework; (2) a multi-source uncertainty prediction model that accurately quantifies different sources of uncertainty through kernel density regularized evidence prediction; (3) an active deep learning model that systematically integrates different types of uncertainty for effective data sampling in a high-dimensional space. Extensive experiments are conducted over both synthetic and real-world data to demonstrate the effectiveness of the proposed ADL model.

## 2 RELATED WORK

**Uncertainty Quantification in Belief/Evidence Theory**: In the belief/evidence theory domain, uncertainty reasoning has been substantially explored such as Fuzzy Logic (De Silva, 2018), Dempster-Shafer Theory (DST) (Sentz et al., 2002), or Subjective Logic (SL) (Jøsang, 2016). Unlike the efforts made in ML/DL above, belief theorists focused on reasoning of inherent uncertainty in information

resulting from unreliable, incomplete, deceptive, and/or conflicting evidence. SL considered uncertainty in subjective opinions in terms of *vacuity* (i.e., lack of evidence) and *vagueness* (i.e., failure of discriminating a belief state) (Jøsang, 2016). Vacuity has been used as an effective vehicle to detect out-of-distribution queries through evidence learning, achieved under the typical DL setting with ample training samples (Sensoy et al., 2018). Recently, other dimensions of uncertainty have been studied, such as *dissonance* (due to conflicting evidence) and *consonance* (due to evidence about composite subsets of state values) (Jøsang et al., 2018).

**Epistemic Uncertainty in Deep Learning**: In DL, *aleatoric* uncertainty (AU) and *epistemic* uncertainty (EU) have been studies using Bayesian Neural Networks (BNNs) for computer vision. AU consists of homoscedastic uncertainty (i.e., constant errors for different inputs) and heteroscedastic uncertainty (i.e., different errors for different inputs) (Gal, 2016). A Bayesian DL (BDL) framework was presented to estimate both AU and DU simultaneously in regression (e.g., depth regression) and classification settings (e.g., semantic segmentation) (Kendall & Gal, 2017). A new type of uncertainty, called *distributional uncertainty*, is defined based on distributional mismatch between the test and training data distributions (Malinin & Gales, 2018).

**Active Learning in Deep Learning**: The common AL methods other than DL-based ones are surveyed in (Settles, 2009). There are limited efforts on actively training DL models for high-dimensional data with a few exceptions. In (Wang & Shang, 2014), an AL model was developed for DL using three metrics for data sampling: least confidence, margin sampling, and entropy. A new approach combines recent advances in BDL into the AL framework to achieve label-efficient DL training (Gal et al., 2017). Another approach advances the AL development by introducing a cost-effective strategy to automatically select and annotate the high-confidence samples, which improves the traditional samples selection strategies (Wang et al., 2016). Data sampling in DL has also been approached as a core-set selection problem (Sener & Savarese, 2018), which requires a large batch to work well. Different from all existing works, the proposed ADL model decomposes the accurately estimated uncertainty into vacuity and dissonance and dynamically balances multi-source uncertainty to achieve active training of DL models with much less labeled data.

## 3 EVIDENCE-AWARE ENTROPY DECOMPOSITION

As discussed earlier, a high entropy may be contributed by difference sources of uncertainty with distinct characteristics. In this section, we conduct a fine-grained theoretical analysis of different types of uncertainty that arise in the context of multi-class problems. The decomposition is conducted under the SL framework, which provides key building blocks for our theoretical analysis.

### 3.1 THEORY OF SUBJECTIVE LOGIC

SL is an uncertain probabilistic logic that is built upon probabilistic logic (PL) (Nilsson, 1986) and belief theory (BT) (Shafer, 1976) while making two unique extensions. First, SL explicitly represents uncertainty by introducing vacuity of evidence (or uncertainty mass) in its opinion representation, which addresses the limitation of using PL to model lack of confidence in probabilities. Second, SL extends the traditional belief function of the BT by incorporating base rates, which serve as the prior probability in Bayesian theory. The Bayesian nature of SL allows it to use second-order uncertainty to express and reason the uncertainty mass, where second-order uncertainty is represented in terms of a probability density function (PDF) over first-order probabilities (Jøsang, 2016). In particular, for multi-class problems, we use a multinomial distribution (first-order uncertainty) to model class probabilities and use a Dirichlet PDF (second-order uncertainty) to model the distribution of class probabilities. Second-order uncertainty enriches uncertainty representation with evidence information, which plays a central role in entropy decomposition as detailed later.

Subjective opinions (or opinions) are the arguments in SL. In the multi-class setting, the subjective opinion of a multinomial random variable y in domain $\mathbb{Y} = \{1, ..., K\}$ is given by a triplet

$$\omega = (\boldsymbol{b}, u, \boldsymbol{a}), \text{ with } \sum_{k=1}^{K} b_k + u = 1 \tag{1}$$

where $\boldsymbol{b} = (b_1, ..., b_K)^T, u$, and $\boldsymbol{a} = (a_1, ..., a_K)^T$ denote the belief mass distribution over $\mathbb{Y}$, uncertainty mass representing vacuity of evidence, and base rate distribution over $\mathbb{Y}$, respectively, and $\forall k, a_k \geq 0, b_k \geq 0, u \geq 0$. The probability that y is assigned to the $k$-class is given by

$$P(\text{y} = k) = b_k + a_k u, \quad \forall k \in \mathbb{Y} \tag{2}$$

which combines the belief mass with the uncertain mass using the base rates. In the multi-class setting, $a_k$ can be regarded as the prior preference over the $k$-th class. When no specific preference is given, we assign all the base rates as $1/K$.

In existing SL literature, there lacks a clear transition between the first order uncertainty given in equation 2 and the second-order uncertainty expressed as a Dirichlet PDF. Here, we make this transition more explicit by introducing a set of random variables $\mathbf{p} = (p_1, ..., p_K)^T$, where $\mathbf{p}$ is distributed on a simplex of dimensionality $K - 1$. We introduce a conditional distribution $P(\mathrm{y} = k|\mathrm{p}_k) = \mathrm{p}_k$, which allows us to represent the marginal distribution in equation 2 by $P(\mathrm{y}) = \int P(\mathrm{y}|\mathbf{p})p(\mathbf{p})d\mathbf{p}$. We define $p(\mathbf{p})$ as a Dirichlet PDF over $\mathbf{p}$: $\mathrm{Dir}(\mathbf{p}|\boldsymbol{\alpha})$, where $\boldsymbol{\alpha} = (\alpha_1, ..., \alpha_K)^T$ is $K$-dimensional strength vector, with $\alpha_k \geq 0$ denoting the effective number of observations of the $k$-th class. SL explicitly introduces the uncertainty evidence through a non-informative weight $W$ and redefine the strength parameter as

$$\alpha_k = r_k + a_k W, \text{ with } r_k \geq 0, \forall k \in \mathbb{Y} \tag{3}$$

where $r_k$ is the amount of evidence (or the number of observations) to support the $k$-th class and $W$ is usually set to $K$, i.e., the number of classes. Given the new definition of the strength parameter, the expectation of the class probabilities $\mathbf{p} = (p_1, ..., p_K)^T$ is given by

$$\mathbb{E}[p_k] = \frac{\alpha_k}{\sum_{j=1}^{K} \alpha_j} = \frac{r_k + a_k W}{\sum_{j=1}^{K} r_j + W} \tag{4}$$

where $a_k = 1/K$. By marginalizing out $\mathbf{p}$, we can derive an evidence-based expression of belief mass and uncertainty mass:

$$b_k = \frac{r_k}{S} \quad \forall k \in \mathbb{Y}, \quad u = \frac{W}{S}, \text{ with } S = \sum_{k=1}^{K} \alpha_k \tag{5}$$

SL categorizes uncertainty into two primary sources (Jøsang, 2016): (1) basic belief uncertainty that results from specific aspects of belief mass in isolation and (2) intra-belief uncertainty that results from the relationships between belief masses and uncertainty mass. Since we focus on the multi-class setting, no composite values (i.e., simultaneously assigned to multiple classes) are allowed. As a result, these two sources of uncertainty boil down to *vacuity* and *dissonance*, respectively, that correspond to vacuous belief and contradicting beliefs. In particular, vacuity of an opinion $vac(\omega)$ is captured by uncertainty mass $u$, which is defined in equation 5 and dissonance of an opinion (Jøsang et al., 2018) is defined as

$$diss(\omega) = \sum_{k=1}^{K} \left( \frac{b_k \sum_{j \neq k} b_j \mathrm{Bal}(b_j, b_k)}{\sum_{j \neq k} b_j} \right), \mathrm{Bal}(b_j, b_k) = \begin{cases} 1 - \frac{|b_j - b_k|}{b_j + b_k} & \text{if } b_i b_j \neq 0 \\ 0 & \text{if } \min(b_i, b_j) = 0 \end{cases} \tag{6}$$

where $\mathrm{Bal}(b_j, b_k)$ is the relative mass balance function between two belief masses. The belief dissonance of an opinion is measured based on how much belief supports individual classes. Consider a binary classification example with a binomial opinion given by $(b_1, b_2, u, \boldsymbol{a}) = (0.49, 0.49, 0.02, \boldsymbol{a})$. Based on equation 6, it has a dissonance value of $0.98$. In this case, although the vacuity is close to zero, a high dissonance indicates that one cannot make a clear decision because both two classes have the same amount of supporting evidence and hence strongly conflict with each other.

## 3.2 Evidence-Based Entropy Decomposition

By leveraging the second-order uncertainty representation, we formally show that the entropy of a predicted class distribution $P(\mathrm{y})$ can be decomposed into vacuity and dissonance. Our major theoretical results indicate that the uncertainty of a high-entropy data sample may be caused by either lack of evidence (i.e., high vacuity) or conflict of strong evidence (i.e., high dissonance) but not both. By clearly identifying the sources of uncertainty instead of using them in a combined form as in entropy, the evidence based decomposition of entropy provides deeper insights on the nature of uncertainty, which provides important guidance for an AL model to more effectively explore a large and high-dimensional search space for efficient data sampling.

**Lemma 1. Dissonance maximization.** *Given a total Dirichlet strength $S = CK$, where $C \geq 1$ and $K$ is the number of classes, for any opinion $\omega$ on a multinomial random variable* y, *we have*

$$\max diss(\omega) = 1 - \frac{1}{C} \tag{7}$$

**Corollary 1.** *The dissonance $diss(\omega)$ is approaching (but not reaching) 1 when all the evidence $r_k$'s are set to equal and $S \to \infty$; it reaches 0 when $S = K$:*

$$\begin{cases} \lim_{S \to \infty} diss(\omega) = 1 & \text{if } r_1 = ...r_k... = r_K \\ diss(\omega) = 0 & \text{if } S = K \end{cases} \tag{8}$$

**Lemma 2. Vacuity maximization.** *For any opinion $\omega$ on a multinomial random variable* y, *we have $0 \leq vac(\omega) \leq 1$ and the maximum vacuity is achieved when $\sum_{k=1}^{K} r_k = 0$.*

**Theorem 1.** *Let* y *denote a multinomial random variable, $\omega_y$ denote its opinion, $S$ denote its total Dirichlet strength, and $H[y]$ be the entropy of* y. *$H[y]$ can be maximized under two different and non-overlapping conditions: (1) for $S = K$ and assuming non-informative base rates, $y^* = \arg\max H[y] \Leftrightarrow y^* = \arg\max vac(\omega_y)$; (2) for $S \to \infty, y^* = \arg\max H[y] \Leftrightarrow y^* = \arg\max diss(\omega_y)$.*

A more intuitive interpretation of the main results in Theorem 1 is as following. A high-entropy data sample supported by a strong evidence (i.e., $S \gg K$) is caused by a high dissonance (i.e., conflict of evidence); a high-entropy data sample supported by little evidence (i.e., $S \approx K$) is caused by a high vacuity (i.e., lack of evidence). Through the second-order uncertainty representation, we offer an evidence based interpretation of entropy that allows us to identify two different sources of uncertainty that both cause a high entropy. The multi-source uncertainty will provide important information to design a fine-grained sampling function for AL, which will be detailed in next section.

## 4 MULTI-SOURCE UNCERTAINTY AWARE ACTIVE DEEP LEARNING

In order to best use the uncertainty information, the ADL model should first be able to provide an accurate uncertainty estimation based on very limited training data. This, coupled with the large number of parameters of the DL model, poses a fundamental challenge due to a higher risk of model overfitting. As shown earlier, inaccurate uncertainty estimation will cause the model to miss labeling important data samples that can help accurately detect the decision boundary if labeled.

In addition, since both vacuity and dissonance are derived from second-order uncertainty, solely predicting the class label or its distribution does not provide sufficient information for multi-source uncertainty prediction. Instead of predicting the class label distribution, the proposed ADL model directly estimates the supporting evidence (i.e., $r_k$'s) for each class, which is a central element that can be used to quantify belief mass and uncertainty mass according to equation 5. To better address overfitting, we develop a novel loss function that augments DL based evidence prediction with uncertainty anchor sample identification through kernel density estimation (KDE). These anchor samples are unlabeled data that inform the ADL which areas of the data space are less explored. Optimizing this loss function will ensure that ADL predicts high vacuity over these areas. Furthermore, through KDE, these less explored areas are automatically ranked based on their data density. This nice property allows the ADL to effectively prioritize the sampling order over these areas and iterative visit them based on their data density. Finally, we introduce our novel sampling function that systematically integrates accurately estimated multi-source uncertainty for active deep learning.

### 4.1 UNCERTAINTY ANCHOR SAMPLE IDENTIFICATION

Let $\mathbb{X}_u$ and $\mathbb{X}_t$ denote the sets of unlabeled candidate and training samples, respectively. The probability density of the two populations with a kernel function $k(\cdot, \cdot)$ can be estimated as follows:

$$p_u(\boldsymbol{x}) = \frac{1}{|\mathbb{X}_u|} \sum_{\boldsymbol{x}_n \in \mathbb{X}_u} k(\boldsymbol{x}, \boldsymbol{x}_n), \quad p_t(\boldsymbol{x}) = \frac{1}{|\mathbb{X}_t|} \sum_{\boldsymbol{x}_n \in \mathbb{X}_t} k(\boldsymbol{x}, \boldsymbol{x}_n) \tag{9}$$

Since we aim to identify unlabeled anchor samples to inform the model areas in the space that are less explored by the training data, these samples should be from areas having a high density mass with respect to $p_u(\boldsymbol{x})$ but low density mass with respect to $p_t(\boldsymbol{x})$. The problem is formalized as:

$$\max_{\mathbb{A} \subseteq \mathbb{X}_u} \lambda \sum_{\boldsymbol{x} \in \mathbb{A}} p_u(\boldsymbol{x}) - \sum_{\boldsymbol{x} \in \mathbb{A}} p_t(\boldsymbol{x}) \tag{10}$$

The first term ensures that the selected area has abundant candidate data points to sample so that it has lower risk of containing isolated noise. The second term makes sure the selected region is located OOD with respect to the current training data. The optimal set for equation 10 is given by:

$$\mathbb{A}^* = \{x | \lambda p_u(\boldsymbol{x}) - p_t(\boldsymbol{x}) > 0\} \tag{11}$$

where $\lambda \in [0, 1]$ is used to control the size of $\mathbb{A}^*$ for given the candidate and training datasets.

### 4.2 Multi-source uncertainty prediction

The set of uncertainty anchor samples $\mathbb{A}^*$ represents areas in the data space that are cohesively distributed far away from the current training data. As these data are essentially OOD with respect to the current training data, their predicted vacuity should be high, which implies low predicted evidence due to Lemma 2. We leverage this information by constructing an evidence strength loss, $\mathcal{L}_{Evi}^{(u)}$, which forces the model to predict low evidence for $\boldsymbol{x}_u \in \mathbb{A}^*$:

$$\mathcal{L}_{Evi}^{(u)}(\mathbb{A}^*, \Theta) = |\mathbf{1}_{(\mathrm{x_u} \in \mathbb{A}^*)}^T f(\boldsymbol{x}_i | \Theta)| \tag{12}$$

where $\mathbf{1}_{(\mathrm{C})} = \mathbf{1}$ if $C$ is true and 0 otherwise; $\boldsymbol{r}_i = f(\boldsymbol{x}_u | \Theta)$ is the output of the DL model, representing the predicted supporting evidence of $\boldsymbol{x}_u$, and $\Theta$ is the set of DL model parameters. Since we require $r_k \geq 0$, an activation layer (i.e., ReLu) is used to replace the softmax layer as commonly used in other NN classifiers. The evidence strength loss is the key component to our proposed overall loss function. Samples in $\mathbb{A}^*$ act as anchors to provide the model a preview of certain areas that out of its current knowledge. The model is guided to put less belief mass on those areas, leading to more accurate uncertainty estimation and eventually benefit the multi-source uncertainty based data sampling. Furthermore, since the activation layer is used for model output, equation 12 essentially performs $l_1$ regularization to last hidden layer's weight matrix and bias vector. We want to emphasize that our approach demands no additional labeling cost. The anchor samples are dynamically detected according to the current training and put into use without their actual label being known.

We proceed to define our overall loss function. For training sample $\boldsymbol{x}_i$, let $\boldsymbol{y}_i$ encode the ground-true class label $k$ by setting $y_{ik} = 1$ and $y_{ij} = 0, \forall j \neq k$. Let $\mathrm{Cat}(\hat{y}_i = k | \mathbf{p}_i(\Theta))$ be the likelihoood, where $\mathbf{p}_i(\Theta) \sim \mathrm{Dir}(\mathbf{p}_i | \boldsymbol{\alpha}_i(\Theta))$ and $\boldsymbol{\alpha}_i(\Theta) = f(\boldsymbol{x}_i | \Theta) + W \boldsymbol{a}_i$. We set the non-informative weight $W = K$ and base rates $a_{ik} = 1/K, \forall k$. The expected sum of squares loss is defined as

$$\mathcal{L}^{(i)}(\Theta) = \mathbb{E}_{\mathbf{p}_i \sim \mathrm{Dir}(\mathbf{p}_i(\Theta)|\boldsymbol{\alpha}_i(\Theta))} ||\boldsymbol{y}_i - \mathbf{p}_i||_2^2 = \sum_{j=1}^{K}(y_{ij}^2 - 2y_{ij}\mathbb{E}[\mathrm{p}_{ij}(\Theta)] + \mathbb{E}[\mathrm{p}_{ij}(\Theta)^2]) \tag{13}$$

Minimizing $\mathcal{L}^{(i)}(\Theta)$ has the effect of jointly minimizing the prediction error and the variance of $\mathbf{p}_i$ (Sensoy et al., 2018), hence reducing the uncertainty. This can be seen using identity $\mathbb{E}[\mathrm{p}_{ij}(\Theta)^2] = \mathbb{E}[\mathrm{p}_{ij}(\Theta)]^2 + \mathrm{Var}(\mathrm{p}_{ij}(\Theta))$ and rearranging the terms on the r.h.s. of equation 13. Our overall loss function is defined as:

$$\sum_{\boldsymbol{x}_i \in \mathbb{X}_t} \mathcal{L}^{(i)}(\Theta) + \lambda_1 \sum_{\boldsymbol{x}_u \in \mathbb{X}_u} \mathcal{L}_{Evi}^{(u)} + \lambda_2 \mathcal{L}_2(\Theta) \tag{14}$$

where $\mathcal{L}_2(\Theta)$ is the standard $L_2$ regularizer of the network parameters.

### 4.3 Data sampling for Active Deep Learning

According to Lemma 2, a data sample's vacuity is maximized when the model assigns zero evidence to all $K$ classes. This indicates the model has never seen a similar data sample from training. Annotating samples with a large vacuity can help the ADL gain most new knowledge of the data space. It has the effect of guiding the model to explore the most important areas, which is especially critical for a high-dimensional data space. In AL, the true decision boundary can be easily skewed due to limited initial training. The vacuity-aware search helps the model fast converge to the true decision boundary without excessively sampling around the wrong one. Moreover, it is also effective to discover new types of classes whose instances have never exposed to the model, as shown in our experiments. According to Lemma 1, a data sample's dissonance is maximized when the model assigns equally high (close to infinity) evidence to all $K$ classes. These strong conflicting evidence received from different classes indicate the data sample is located near the decision boundary where multiple classes are heavily overlapped. Annotating samples with high dissonance helps the model further fine-tune the decision boundary, leading to better discriminative power.

We design a sample function that best leverages these two important and complementary sources of uncertainty to most effectively guide ADL. Intuitively, we would like ADL to rely more on vacuity in the early phase of AL, which can most effectively shape the decision boundary and avoid fine-tuning the wrong decision areas. As AL goes, dissonance should gradually gain a higher weight, which allows ADL to further fine-tune the decision boundary that has the right shape but is less accurate, aiming to maximize the discriminate power of the model. The sample function is given:

$$\boldsymbol{x}^* = \arg\max_{\boldsymbol{x} \in \mathbb{X}_u}[diss(\omega(\boldsymbol{x})) + \beta vac(\omega(\boldsymbol{x})] \tag{15}$$

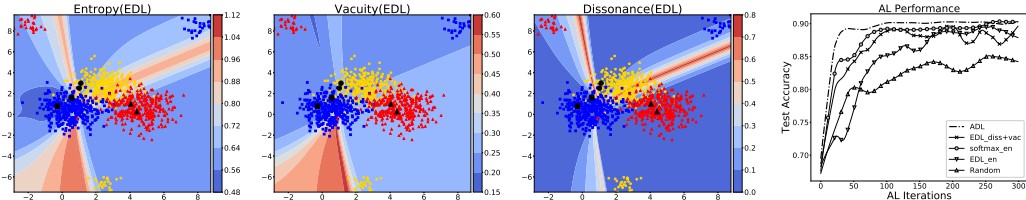

Figure 2: Uncertainty prediction result from EDL and AL performance comparison;

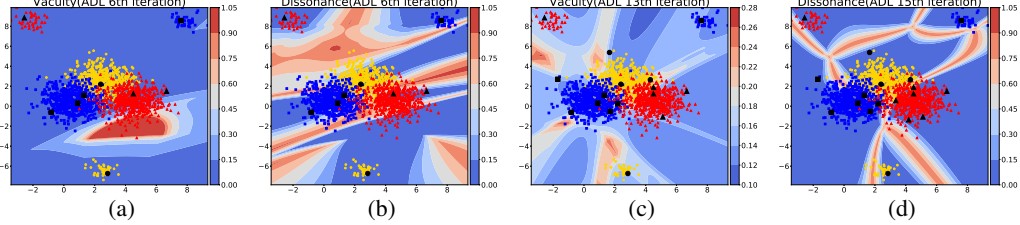

Figure 3: (a) Vacuity and (b) Dissonance of iteration 6 when ADL first discovers all hidden OOD areas; (c) Vacuity of iteration 13 when ADL starts penalizing vacuity in the sampling function; (d) Dissonance after two iterations of penalizing vacuity.

where $\beta$ is an annealing coefficient to gradually balance between vacuity and dissonance based on the rationale given above. More specifically, the importance of vacuity reduces as there are less "vacuous" areas in the data space w.r.t. the current training data. This implies that the training data can well approximate the entire data space. Thus, one natural way to quantify $\beta$ is to use the inverse of mutual information $KL(p_{u,t}||p_t)$, where the joint density distribution $p_{u,t}$ can be estimated as $p_{u,t}(\boldsymbol{x}) = \frac{1}{|\mathbb{X}_u|+|\mathbb{X}_t|} \sum_{\boldsymbol{x}_n \in \mathbb{X}_u \cup \mathbb{X}_t} k(\boldsymbol{x}, \boldsymbol{x}_n)$. In practice, calculating the mutual information for each AL iteration is expensive. We use a heuristic surrogate: min_max=$\min_{\mathbf{x}_u \in X_u} \max_{\mathbf{x}_t \in X_t} k(\boldsymbol{x}_t, \boldsymbol{x}_u)$ and set $\beta = 1 - dT$ if min_max does not change within the past few AL iterations, where $T$ denotes the current iteration of AL and $d$ is a fixed decay rate (set to $1/100K$ in our experiments).

## 5 EXPERIMENTS

In this section, we report our experimental results on both synthetic and real-world data. The synthetic experiment aims to verify the key theoretical properties of ADL, including entropy decomposition and multi-source uncertainty prediction, and how these properties contribute to AL. The real-data experiment aims to compare ADL and its competitors. We focus on testing in classical AL environment, where the initial training set only includes limited samples from some classes with samples from other classes completely missing. In each AL iteration, we sample one or a small batch of data instances. This is fundamentally different than some recent DL based AL methods, such as (Sener & Savarese, 2018), which perform batch-mode sampling with a large batch size (larger than our entire labeled samples). All models uses the same DL architecture. For synthetic data, we adopt a 3-layer MLP with tanh for activation. For real data, we use LeNet with Relu for activation.

### 5.1 SYNTHETIC DATA

The synthetic experiment is designed to show: (1) whether ADL accurately captures different sources of uncertainty, and (2) whether accurately estimated uncertainty leads to better AL behavior. To mimic the existence of OOD, we generate three mixtures of Gaussian's. Each mixture consists of a major and a smaller (i.e., OOD) clusters with 750 and 50 samples, respectively. We center the major Gaussian components from each class in the middle of the feature space and put their corresponding OOD components away from them. In Figure 1, we show that a classical DL model with a softmax layer provides very inaccurate uncertainty estimation. In contrast, the proposed ADL model not only provides accurate entropy prediction but also successfully decomposes it into vacuity and dissonance. Figure 2 shows the uncertain prediction result from EDL (Sentz et al., 2002), which can also provide evidence prediction but requires ample training data. Suffering from insufficient training, EDL is inaccurate in its entropy prediction, especially for the OOD clusters. While EDL does not provide entropy decomposition, we use its predicted evidence to compute vacuity and dissonance as shown in Figure 2. However, neither of them is accurately predicted as low vacuity is predicted for the three OOD clusters where there is no training data and high dissonance is predicted in areas with no nearby training data to show conflicting evidence.

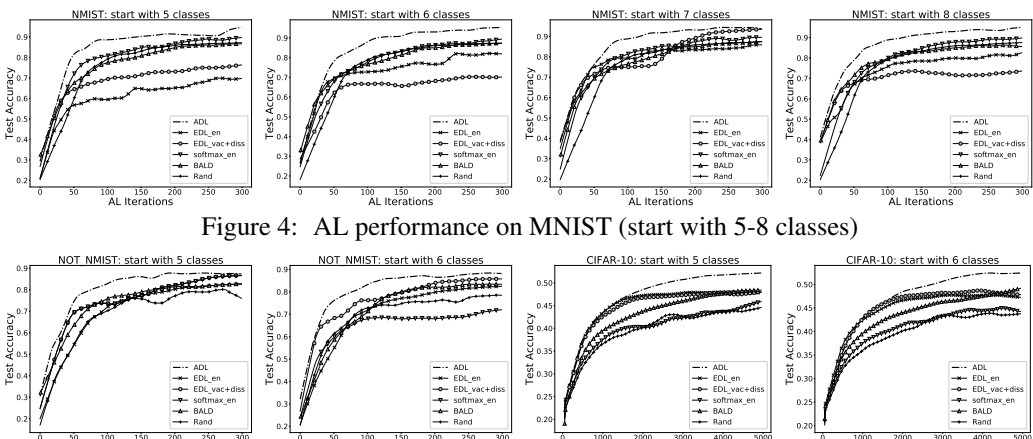

Figure 4: AL performance on MNIST (start with 5-8 classes)

Figure 5: AL performance on notMNIST and CIFAR-10 (start with 5 and 6 classes; 7,8 in Appendix)

Figure 3 shows the first time when ADL selects at least one data sample from each OOD area, high vacuity is assigned to an area with no training data but many unlabeled data. Meanwhile, high dissonance indicates that refining the decision boundary may be more instrumental to improve the model performance. A few iterations later, ADL starts to penalize vacuity. While vacuity is still accurately estimated (high in vacuous regions), it becomes less useful for sampling (since very few unlabeled data is located nearby). Two iterations later, penalizing on vacuity helps to choose data samples that significantly refine the decision boundary. The superior AL performance of ADL as shown in Figure 3 further confirms the effectiveness of ADL's key properties as demonstrated above.

## 5.2 REAL DATA

The real-world experiment is conducted on three datasets, MNIST, notMNIST, and CIFAR-10, all of which have ten classes. To mimic the real-world AL scenario, we leave 2-5 classes out for initial training and there are 5 labeled samples for each available class. A good AL model is expected to discover samples of unknown classes in an early stage to effectively improve model accuracy. We compare the proposed model with **EDL** (Sensoy et al., 2018) (entropy, vacuity+dissonance), **BALD** (Gal et al., 2017) (epistemic), and **softmax** (entropy, random), where in the parenthesis are the uncertainty measurements used for sampling. Figures 4 and 5 show that ADL consistently outperforms other models on all three datasets. The advantages of ADL are twofold. First, entropy decomposition gives ADL flexibility to meet distinct sampling need at different AL phases. In an early stage, the fast accuracy improvement is achieved by the vacuity guided sampling where the most *representative* samples are labeled with high priority. Gradually, ADL switches to dissonance guided sampling to refine the decision boundary by labeling the most *informative* samples to improve its discriminative power. In contrast, sampling methods utilizing a unified uncertainty (e.g., epistemic uncertainty and entropy) lack such flexibility to adjust the sampling behavior, leading to either slow convergence or lower model accuracy. Second, compared with EDL, which can also perform evidence prediction, ADL is superior due to accurate uncertainty estimation using the effective loss function. For both synthetic and real data, we observe that ADL identifies samples from missing classes at least around 20% faster than using EDL and other models.

## 6 CONCLUSION

We present a novel active deep learning model that systematically leverages two distinct sources of uncertainty, vacuity and dissonance, to effectively explore a large and high-dimensional data space for label-efficient training of DL models. The proposed ADL model benefits from the evidence-based entropy decomposition that follows from our theoretical analysis of belief vacuity and belief dissonance under the SL framework. The multi-source uncertainty can be accurately estimated through a novel loss function that augments DL based evidence prediction with vacuity-aware regularization of the model parameters. Through dynamically balancing the importance of vacuity and dissonance, a sampling function is designed to first explore the critical areas of the data space and then fine-tune the decision boundary to maximize its discriminate power. Extensive experiments conducted over both synthetic and real data help verify the theoretical properties and empirical performance of the proposed ADL model.

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

## A APPENDIX

PROOF OF LEMMA 1

*Proof.* Let $B_{jk}$ denote $\text{Bal}(b_j, b_k)$. Since $0 \leq b_k \leq 1$ (as $S = \sum_k b_k + K$), we have $0 \leq B_{jk} \leq 1$. In addition, $B_{jk} = 1$, if $b_j = b_k \neq 0$; $B_{jk} = 0$, if $b_j b_k = 0$. Thus, we have $\sum_{j \neq k} b_j B_{jk} \leq \sum_{j \neq k} b_j$, where the equality holds when $B_{jk} = 1, \forall j$. Therefore, we have

$$diss(\omega) = \sum_{k=1}^{K} b_k \left[ \frac{\sum_{j \neq k} b_j B_{jk}}{\sum_{j \neq k} b_j} \right] \leq \sum_{k=1}^{K} b_k \overset{(a)}{=} \frac{\sum_{k=1}^{K} r_k}{S} \overset{(b)}{=} \frac{S - K}{S} = 1 - \frac{1}{C} \quad (16)$$

where (a) is due to the definition of $b_k$ in equation 5 and (b) is due to the summation constraint in equation 1 and $W = K$. $\square$

PROOF OF LEMMA 2

*Proof.* Using the definition of uncertainty mass in equation 5 and substituting $W$ by $K$, we have

$$0 \leq vac(\omega) = \frac{K}{S} = \frac{K}{\sum_{k=1} r_k + K} \leq 1 \quad (17)$$

where equality is achieved when $\sum_{k=1}^{K} r_k = 0$. $\square$

PROOF OF THEOREM 1

*Proof.* For (1), ($\Rightarrow$) is easy to show as $S = K$ implies $\sum_{k=1}^{K} r_k = 0$ and $vac(\omega_{y^*}) = 1$; for ($\Leftarrow$), using equation 2 and non-informative base rates, we have $P(y^* = k) = 1/K, \forall k$, which achieves a maximum $H[y^*]$ as $\log K$.

For (2), we first prove ($\Rightarrow$). For $y^* = \arg\max H[y]$, we have $P(y^* = k) = 1/K, \forall k$. Thus, $(r_k + a_k K)/S = 1/K, \forall k$. For $S \to \infty$, denote $S = CK$ and we have $r_k/S + a_k/C = 1/K, \forall k$. Let $C \to \infty$, we have $r_k/S \to 1/K, \forall k$. Thus, we have $y^* = \arg\max diss(\omega_y)$ due to Corollary 1. To prove ($\Leftarrow$), $diss(\omega_{y^*}) = 1$ implies that $r_1 = ... r_k ... = r_K$ and $S \to \infty$. Hence, $\lim_{S \to \infty} P(y^* = k) = \lim_{S \to \infty} (r_k + a_k K)/S = 1/K$, which implies that $y^* = \arg\max H[y]$ for $S \to \infty$. $\square$

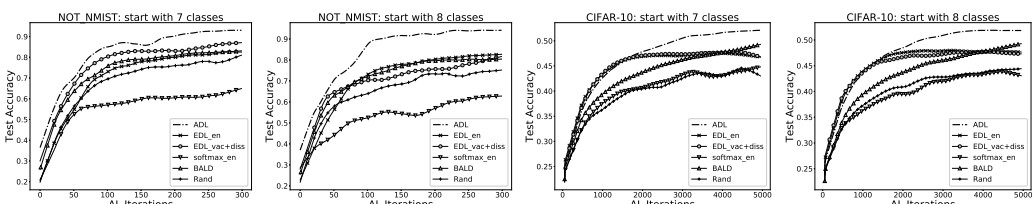

Figure 6: AL performance on notMNIST and CIFAR-10 (start with 7 and 8 classes)

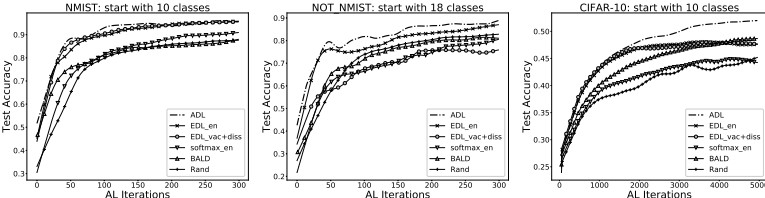

Figure 7: AL performance with no missing classes

ADDITIONAL EXPERIMENTAL RESULTS

We obtain similar AL curves for notMNIST and CIFAR-10 when starting AL with 7 and 8 classes as with 5 and 6 classes, which are shown in Figure 6. In Figure 7, we also report the AL performance on the three datasets when there is no missing class. ADL still achieves the best performance in all cases with slightly less advantage than other models.

EXPERIMENTAL SETTINGS

We choose the Adam optimizer to train ADL for 600 epochs and setting the learning rate to 0.001. The coefficient of evidence strength loss, $\lambda_1$, is set to 0.005 (cross validated from $\{0.001, 0.005, 0.03, 0.05\}$). The coefficient of the $L_2$ regularizer, $\lambda_2$, is set to 0.05 (cross validated from $\{0.001, 0.005, 0.01, 0.03, 0.05, 0.08\}$). The $\lambda$ for anchor sample identification in equation 10 is set to 0.005 (cross validated $\{0.001, 0.005, 0.03, 0.05\}$). We choose RBF as our kernel function with length scale set to 1 (cross validated from $\{0.01, 0.1, 1\}$).

EFFECTIVENESS OF KDE BASED UNCERTAINTY ANCHOR SAMPLE IDENTIFICATION

In this section, we conduct additional experiment to evaluate the effectiveness of the KDE based uncertainty anchor sample identification method. Uncertainty anchor sample identification is an integral component of the ADL model, which aims to guide the model to be uncertain in the OOD areas with respect to the current training data (instead of providing the final prediction). Therefore, other more advanced kernel functions/similarity measures that are specifically designed for high-dimensional data can be used for the same purpose without affecting the overall model. However, when choosing a specific technique, it is also important to consider both the quality of the data samples and efficiency as fast identification of these data samples is critical for AL which is usually performed in real time. Since the model is constantly changing as it continues to explore the data space, new uncertainty data samples need to be discovered in each AL iteration. We have conducted three additional experiments to demonstrate which technique can achieve such a good balance.

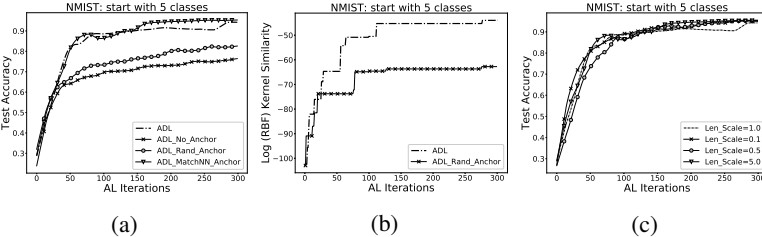

(a)                              (b)                              (c)

Figure 8: Effectiveness of KDE based anchor sample identification. (a) Comparison with random selection, no anchor samples, and attention kernel; (b) Comparison of min-max similarity of different techniques; (c) Impact of the characteristic length scale

- We have compared KDE with the randomly selected anchor samples from unlabeled data and not using any anchor samples in Figure 8(a). KDE clearly outperforms random selection, which in turn performs better than not using any anchor samples. We further confirm the positive result by evaluating the min-max similarity between the unlabeled and training data. If KDE is able to identify anchor samples from the desired OOD regions of the feature space (although the estimated density in that region may not be very accurate), the sampling process would be guided correctly and the min-max similarity would increase in the next AL iteration as the result. Figure 8(b) compares the min-max similarity of KDE with random selection. The result shows that with KDE, the model covers the unlabeled feature space much more efficiently as AL moves forward.

- We have adopted the attention kernel as a more advanced distance metric to replace the RBF kernel in the proposed anchor sample identification component. The attention kernel is the major component in the matching network (Vinyals et al., 2016), where the spatial invariance is ensured by CNN and the dimensionality of the inputs is reduced through two correlated LSTM projections. However, the attention kernel (our current implementation) is much slower to compute as compared with KDE especially when facing a very large unlabeled pool as the entire candidate data samples need to be embedded every iteration when the training/testing data are changed along with AL. Thus, if the improvement is not significant (see Figure 8(a)) and when the efficiency becomes a bottleneck for a large unlabeled pool, the proposed KDE approach appears to be a good choice as it can provide a good balance between quality and efficiency, which is critical for AL.

- We have investigated the impact of the characteristic length scale used in RBF kernel on AL performance. Figure 8(c) shows that the ADL model performance is fairly robust to the length scale and only shows minor change with different choices.

ABLATION STUDY

We have conducted a detailed ablation study to clearly demonstrate the effectiveness of each major technical components:

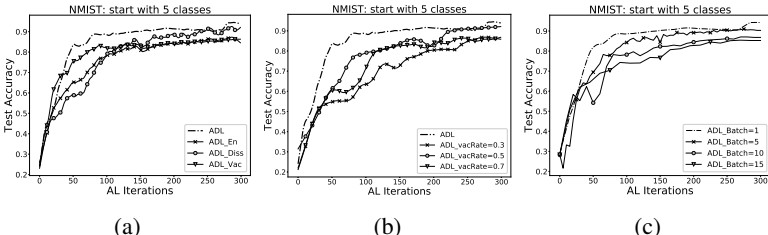

(a)                          (b)                          (c)

Figure 9: Ablation study and batch-model AL. (a) Comparison of different sampling criteria; (b) Comparison with different fixed vacuity/dissonance weighting; (c) Batch-model AL performance.

- Figure 9(a) compares proposed sampling method with other different sampling criteria: entropy, vacuity only, and dissonance only. The result confirms the effectiveness of the dynamically balanced sampling method. It is interesting to see that using vacuity alone performs quite well in the initial phase but only converges to a lower accuracy in the end. In contrast, using dissonance is slow to start but able to converge to a higher accuracy. The entropy curve roughly stays in the middle of the above two curves.

- The effectiveness of using the anchor samples has already been demonstrated in Figure 8(a) by comparing ADL with not using any anchor samples and randomly selected anchor samples from unlabeled data.

- Figure 9(b) shows the results using different vacuity/dissonance ratios but keeping fixed throughout the AL process. The dynamically balanced sampling method clearly outperforms the fixed weighting. This also demonstrates the usefulness of the proposed entropy decomposition theory. Since the sampling goal of AL changes with the accumulation of the labeled data, the optimal AL behavior can only be achieved by adaptively adjust the importance of vacuity and dissonance in the sampling function.

Finally, we have conducted batch-mode AL and reported the results in Figure 9(c). As can be seen, as the batch size increases, the performance decreases. This is expected as there is no special strategy to diversify the samples chosen in the same batch. We will leave this to our future work as the current model is not designed specifically for batch mode AL.

SAMPLE IMAGES CHOSEN BY ADL

In this section, we have visualized the image samples selected by ADL in the early and later stages of active learning to help better understand the role of vacuity and dissonance in data sampling. In order to better demonstrate the effectiveness of the vacuity measurement, we start active learning with 5 classes omitted from the initial training. Later we will see how does high vacuity at the early stage of active learning helps fast identify missing classes. Figure 10 shows the samples with highest vacuity selected by ADL in the first 30 AL iterations. The first four of them are from missing classes. This clearly demonstrates the effectiveness of using vacuity to explore the data space. As a result, data samples from the missing classes are quickly identified and being labeled. The last sample is from class '3', whose examples have already been exposed to ADL. However, the writing style of this sample is very different than other instances from the same class, which result in a high vacuity.

Figure 11 shows the samples with highest dissonance selected by ADL in the last 100 AL iterations. By observing their predicted belief mass, we find that the high dissonance result is due to the conflicting belief over multiple classes. For example, the first sample is confusing between classes '4' and '6'; the second sample is confusing among classes '5','6', and '8'; the third sample is confusing

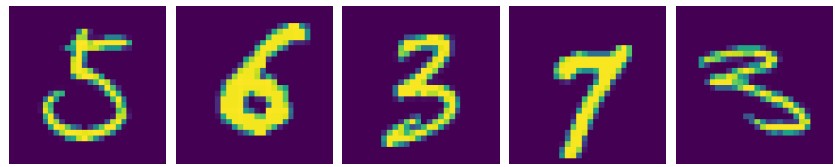

Figure 10: Samples with a high vacuity in early AL iterations

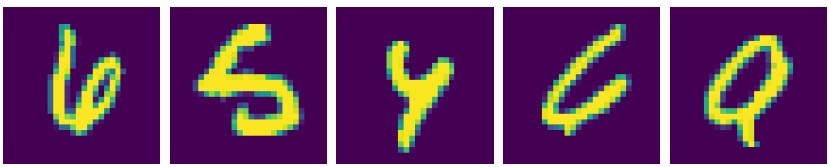

Figure 11: Samples with a high dissonance in late AL iterations

among classes '4','7', and '9'; the fourth sample is confusing between classes '4' and '6'; and the Fifth sample is confusing between classes '0' and '9'.

SOURCE CODE

The code for this work can be found in `https://drive.google.com/drive/folders/1imwnOahh8HtHK_g_HSTb4TCxZ7YG04ay?usp=sharing`

