# OpenReview forum: "Evidence-Aware Entropy Decomposition For  Active Deep Learning"
_ICLR.cc/2020/Conference — Reject_

### Official Review · AnonReviewer1 · 2019-10-24
**Official Blind Review #1**

**Rating:** 6

**Review:**

This paper propose an active deep learning model. By leveraging subjective Logic, they propose to decompose the entropy of a predicted class distribution into vacuity (lack of evidence) and dissonance (conflict of strong evidence). Instead of using the predicted class distribution, they estimate the supporting evidence for each class. In the actual data sampling stage, they first sample from those high-vacuity dense region, to shape the true decision boundary, and then gradually sample from those high-dissonance region to fine-tune the decision boundary. They show better performance than the baselines on both synthetic and real datasets.

First of all, for the readers who are not familiar with Subjective Logic or probabilistic logic in general, it is a bit hard to follow the reasoning behind the  equations in Sec 3 and  4. Specifically, what is the intuition behind the dissonance of an opinion in Eq (6)? What is the advantage of using Subjective logic framework.

In the experiments on MNIST, it would be best to visualize the image samples selected by the active learning model in the early and later stages, so that we can have a more intuitive understanding of vacuity vs. dissonance.

**Experience Assessment:**

I do not know much about this area.

**Review Assessment: Checking Correctness Of Derivations And Theory:**

I assessed the sensibility of the derivations and theory.

**Review Assessment: Checking Correctness Of Experiments:**

I assessed the sensibility of the experiments.

**Review Assessment: Thoroughness In Paper Reading:**

I made a quick assessment of this paper.

---

> ### Author Response · Authors · 2019-11-11
> **Response to Reviewer #1**
>
> We thank the reviewer for the thoughtful feedback and recommendation for acceptance.
>
> Q1. The intuition of dissonance and connection to SL.
>
> The definitions of vacuity and dissonance (equation 6) are given by subjective logic, which has been made clear in the revised paper. Intuitively, the dissonance is computed by collecting the conflicting evidence from all $K$ classes, where each class is weighted by its corresponding belief mass $b_k$. To compute the conflict evidence for class $k$, we first compute the relative mass balance between $b_k$ and $b_j$ from each of the other classes. The relative mass balance has its maximum at 1 when $b_j=b_k$, showing highest conflict and it has the minimum at 0 when one of the belief masses equals 0. Then, a weighted average is computed overall all the $K-1$ classes (except for class $k$), which is used to denote the total conflicting evidence for class $k$. Finally, a weighted sum of the conflicting evidence for all classes are computed to give the dissonance. We have also added one example to further illustrate the importance of dissonance (see page 4 of the revised paper).
>
> Subjective Logic (SL) offers two key uncertainty modeling components that are essential to both our entropy decomposition theory and the design of the sampling function for active deep learning. First, SL explicitly represents uncertainty by introducing vacuity of evidence (or uncertainty mass) in its opinion representation (see equation 1 on page 3). Most DL and other classification models only consider the belief mass $b_k$ through softmax or probabilistic modeling but ignores the vacuity mass $u$. The vacuity mass offers us a critical component to model the vacuous belief where the model uncertainty is caused by the lack of evidence. Second, it employs the second-order uncertainty to enrich the uncertainty representation with the evidence information. The evidence offers deeper insights on the different causes of uncertainty, which allows us to differentiate lack of evidence and conflict of evidence, both of which lead to a high entropy. Thus, the evidence information and second order uncertainty play a central role in our entropy decomposition theory. Furthermore, our paper also fills out a critical gap in the current SL literature, which lacks a clear transition between the first order uncertainty of a multinomial opinion (equation 2 on page 3) and the evidence-based expression of belief and uncertainty mass derived from the second order uncertainty (equation 5 on page 4). In the current SL literature, this relationship is specified by a mapping, which is only vaguely described without a mathematical justification. We address this issue by explicitly introducing the Dirichlet distributed parameters $p_k$’s along with the conditional distribution $P(y=k|p_k)=p_k$. By marginalizing out $p_k$’s, we obtain the first order uncertainty of a multinomial opinion with the evidence-based expression given in equation 5.
>
> Q2. Image samples to show effectiveness of vacuity and dissonance.
>
> Following the reviewer’s suggestion, we have visualized the image samples selected by ADL in the early and later stages of active learning to help better understand the role of vacuity and dissonance in data sampling.
> •	Figure 10 (on page 13) shows the samples with the highest vacuity selected by ADL in the first 30 AL iterations. The first four of them are from missing classes. This clearly demonstrates the effectiveness of using vacuity to explore the data space. As a result, data samples from the missing classes are quickly identified and being labeled. The last sample is from class '3', whose examples have already been exposed to ADL. However, the writing style of this sample is very different than other instances from the same class, which result in a high vacuity. Labeling this data sample will be helpful to classify digits in a similar writing style.
>
> •	Figure 11 (on page 13) shows the samples with the highest dissonance selected by ADL in the last 100 AL iterations. By observing the predicted belief mass, we find that the high dissonance result is due to the conflicting belief mass over multiple classes. For example, the first sample is confusing between classes '4' and '6'; the second sample is confusing among classes '5','6', and '8'; the third sample is confusing among classes '4','7', and '9'; the fourth sample is confusing between classes '4' and '6'; and the last sample is confusing between classes '0' and '9'.

---

### Official Review · AnonReviewer3 · 2019-10-25
**Official Blind Review #3**

**Rating:** 3

**Review:**

The authors consider active deep learning. They propose decomposing predictive entropy into a) vacuity (lack of evidence) and b) dissonance (contradictory evidence). They frame this in terms of "subjective logic". In practice this is achieved by having the NN output the parameters of a Dirichlet, which allows an additional degree of freedom describing variance/vacuity. Dissonance is defined in terms of the support of contradictory classes. To get improved estimates of vacuity they augment the loss with a term regularizing the Dirichlet parameters to be small (low precision) at unlabelled points with higher KDE(unlablled points) than KDE(labelled points). They propose initially weighting vacuity and later dissonance as AL proceeds. Encouraging results are presented on simulated 2D data, MNIST and CIFAR10.

I think the basic idea of separating vacuity and dissonance is interesting, and the demonstration of the failings of existing ADL approaches is valuable. It wasn't clear to me how this "subjective logic" theory gets you to the specific definitions of vacuity/dissonance, or whether these were just proposed by the authors. Equation 6 seems to come out of nowhere (whereas the rest of the derivations using the Dirichlet are very intuitive).

The idea of encouraging the network to be uncertain far from data is also reasonable. While some Bayesian models such as Gaussian process regression with a RBF kernel give you this for free, it is certainly true that DL methods do not have this characteristic in general. Regularizing the r to be small seems like a reasonable way to do this, but I'm not convinced by the kernel density estimate part. DNNs can operate on very high-dimensional, structured inputs. Even the simplest of these, images, requires some degree of spatial invariance (achieved using convolutions) to obtain meaningful predictions. I find it very hard to believe a KDE can do anything meaningful in such spaces, even if you could find a sensible bandwidth (which isn't discussed at all). It is possible of course that random selection of unlabelled points to regularize in this way would work just as well. Unfortunately no ablation study is performed, so we don't know what the individual contribution of the three proposals (moving from vacuity to dissonance, augmented loss and Dirichlet likelihood) is.

How sensitive is the method to the vacuity/dissonance weighting?

The improvement over competing methods for MNIST and CIFAR10 appears to mostly manifest after the initial 20 or so acquistions.

Do these results extend to batch AL? For many applications that's more important.

Overall I thought this paper had some promising ideas but they need to be more thoroughly tested empirically to give some sense of how robust and generalizable the approach is.

**Experience Assessment:**

I have read many papers in this area.

**Review Assessment: Checking Correctness Of Derivations And Theory:**

I assessed the sensibility of the derivations and theory.

**Review Assessment: Checking Correctness Of Experiments:**

I assessed the sensibility of the experiments.

**Review Assessment: Thoroughness In Paper Reading:**

I read the paper thoroughly.

---

> ### Author Response · Authors · 2019-11-11
> **Response to Reviewer #3 [part 3]**
>
> Q3. Ablation study.
>
> We have conducted a detailed ablation study to clearly demonstrate the effectiveness of each major technical component:
>
> 1.	Figure 9(a) (on page 12) compares the proposed sampling method with other different sampling criteria: entropy, vacuity only, and dissonance only. The result confirms the effectiveness of the dynamically balanced sampling method. It is interesting to see that using vacuity alone performs quite well in the initial phase but only converges to a lower accuracy in the end. In contrast, using dissonance is slow to start but able to converge to a higher accuracy. The entropy curve roughly stays in the middle of the above two curves.
> 2.	The effectiveness of using the anchor samples has already been demonstrated in Figure 8(a) by comparing ADL with not using any anchor samples and randomly selected anchor samples from unlabeled data.
> 3.	Figure 9(b) (on page 12) shows the results using different vacuity/dissonance ratios but keeping fixed throughout the AL process. The dynamically balanced sampling method clearly outperforms the fixed weighting. This also demonstrates the usefulness of the proposed entropy decomposition theory. Since the sampling goal of AL changes with the accumulation of the labeled data, the optimal AL behavior can only be achieved by adaptively adjusting the importance of vacuity and dissonance in the sampling function.
>
> Q4. Improvement over competing methods appears to be after the initial 20 acquisitions.
>
> We use a very small number of labeled samples (5 samples/class with samples from certain classes completely missing) to initialize AL. Given the size of the DL model, it needs to acquire a reasonable number of labeled samples to learn a good set of parameters. After the model can gain some accuracy (i.e., after 20 or so iterations), it can start to perform more accurate data sampling and gain performance advantage over other models.
>
> Q5. Extension to batch AL.
>
> We have conducted batch mode AL and reported the results in Figure 9(c) (on page 12). As can be seen, as the batch size increases, the performance decreases. This is expected as there is no special strategy to diversify the samples chosen in the same batch. We will leave this to our future work as the current model is not designed specifically for batch mode AL.

---

> ### Author Response · Authors · 2019-11-11
> **Response to Reviewer #3 [part 2]**
>
> Q2. Effectiveness of KDE.
>
> We agree with the reviewer that KDE may not be able to accurately approximate the density of the data especially in a high dimensional space. Key properties, such as spatial invariance, are important to provide a more accurate prediction. However, we would like to clarify that the role of KDE in our model is to identify the interesting and under-explored areas in the data space to inform the model to be uncertain by estimating a high vacuity in those areas. The AL model only needs a representative data sample from those regions that are OOD with respect to the current training data but which specific data sample to choose is less important. This allows the ADL to sample from these areas for more effective exploration of the data space. It is important to note that the final prediction is still through the deep learning model, which is designed to handle high-dimensional data well.
>
> Furthermore, the uncertainty anchor sample identification is an integral component of the ADL model, which aims to guide the model to be uncertain in the OOD areas with respect to the current training data (instead of providing the final prediction). Therefore, other more advanced kernel functions/similarity measures that are specifically designed for high-dimensional data can be used for the same purpose without affecting the overall model. However, when choosing a specific technique, it is also important to consider both the quality of the data samples and efficiency as fast identification of these data samples is critical for AL which is usually performed in real time. Since the model is constantly changing as it continues to explore the data space, new uncertainty data samples need to be discovered in each AL iteration. We have conducted three additional experiments to demonstrate which technique can achieve such a good balance.
>
> 1.	We have compared KDE with the randomly selected anchor samples from unlabeled data (as suggested by the reviewer) and not using any anchor samples in Figure 8(a) (on page 11). KDE clearly outperforms random selection, which in turn performs better than not using any anchor samples. We further confirm the positive result by evaluating the min-max similarity between the unlabeled and training data. If KDE is able to identify anchor samples from the desired OOD regions of the feature space (although the estimated density in that region may not be very accurate), the sampling process would be guided correctly and the min-max similarity would increase in the next AL iteration as the result. Figure 8(b) (on page 11) compares the min-max similarity of KDE with random selection. The result shows that with KDE, the model covers the unlabeled feature space much more efficiently as AL moves forward.
> 2.	We have adopted the attention kernel as a more advanced distance metric to replace the RBF kernel in the proposed anchor sample identification component. The result is shown in Figure 8(a). The attention kernel is the major component in the matching network [1], where the spatial invariance is ensured by CNN and the dimensionality of the inputs is reduced through two correlated LSTM projections. However, the attention kernel (our current implementation) is much slower to compute as compared with KDE especially when facing a very large unlabeled pool as the entire candidate data samples need to be embedded every iteration when the training/testing data are changed along with active learning. Thus, if the improvement is not significant (see Figure 8(a)) and when the efficiency becomes a bottleneck for a large unlabeled pool, the proposed KDE approach appears to be a good choice as it can provide a good balance between quality and efficiency, which is critical for AL.
> 3.	We have investigated the impact of the characteristic length scale used in RBF kernel on AL performance. Figure 8(c) (on page 11) shows that the ADL model performance is fairly robust to the length scale and only shows minor change with different choices.
>
> [1] Vinyals, Oriol, et al. "Matching networks for one shot learning." Advances in neural information processing systems. 2016.

---

> ### Author Response · Authors · 2019-11-11
> **Response to Reviewer #3 [part 1]**
>
> We thank the reviewer for the constructive feedback. We appreciate your positive evaluation that the key idea presented in our paper is valuable and promising, and our experimental results are encouraging.
>
> Q1. Definitions of vacuity/dissonance.
>
> The definitions of vacuity and dissonance (equation 6) are given by subjective logic, which has been made clear in the revised paper. Intuitively, the dissonance is computed by collecting the conflicting evidence from all $K$ classes, where each class is weighted by its corresponding belief mass $b_k$. To compute the conflict evidence for class $k$, we first compute the relative mass balance between $b_k$ and $b_j$ from each of the other classes. The relative mass balance has its maximum at 1 when $b_j=b_k$, showing highest conflict and it has the minimum at 0 when one of the belief masses equals 0. Then, a weighted average is computed overall all the $K-1$ classes (except for class $k$), which is used to denote the total conflicting evidence for class $k$. Finally, a weighted sum of the conflicting evidence for all classes are computed to give the dissonance. We have also added one example to further illustrate the importance of dissonance (see page 4 of the revised paper).
>
> Subjective Logic (SL) offers two key uncertainty modeling components that are essential to both our entropy decomposition theory and the design of the sampling function for active deep learning. First, SL explicitly represents uncertainty by introducing vacuity of evidence (or uncertainty mass) in its opinion representation (see equation 1 on page 3). Most DL and other classification models only consider the belief mass $b_k$ through softmax or probabilistic modeling but ignores the vacuity mass $u$. The vacuity mass offers us a critical component to model the vacuous belief where the model uncertainty is caused by the lack of evidence. Second, it employs the second-order uncertainty to enrich the uncertainty representation with the evidence information. The evidence offers deeper insights on the different causes of uncertainty, which allows us to differentiate lack of evidence and conflict of evidence, both of which lead to a high entropy. Thus, the evidence information and second order uncertainty play a central role in our entropy decomposition theory. Furthermore, our paper also fills out a critical gap in the current SL literature, which lacks a clear transition between the first order uncertainty of a multinomial opinion (equation 2 on page 3) and the evidence-based expression of belief and uncertainty mass derived from the second order uncertainty (equation 5 on page 4). In the current SL literature, this relationship is specified by a mapping, which is only vaguely described without a mathematical justification. We address this issue by explicitly introducing the Dirichlet distributed parameters $p_k$’s along with the conditional distribution $P(y=k|p_k)=p_k$. By marginalizing out $p_k$’s, we obtain the first order uncertainty of a multinomial opinion with the evidence-based expression given in equation 5.

---

### Author Response · Authors · 2019-11-11
**Response to all reviewers**

We want to first thank all the reviewers for taking time to review our paper and providing their insightful comments and suggestions. We have revised the paper by following the reviewers’ suggestions and the major changes are highlighted in blue in the revised paper uploaded to the review system. They are also summarized below:

1.	We have clarified the definitions of some key concepts introduced in the paper, including vacuity and dissonance. Both the reference and intuitions of the equation 6 are added on page 4. Subjective Logic (SL) offers two key uncertainty modeling components that are essential to both our entropy decomposition theory and the design of the sampling function for active deep learning, including (1) explicitly representing uncertainty by introducing vacuity of evidence (or uncertainty mass) in its opinion representation, and (2) using the second-order uncertainty to enrich the uncertainty representation with the evidence information. The definitions of both vacuity and dissonance were proposed and validated in the belief/evidence theory domain.
2.	We agree that KDE may not obtain very accurate density estimation on very high-dimensional, structured inputs. To clarify this concern, we have thoroughly investigated the effectiveness of using KDE to identify the interesting and under-explored areas in the data space, aiming to guide the model to be uncertain by estimating a high vacuity in these areas. Instead of making a prediction, the KDE based anchor sample identification is to guide the ADL to sample from these areas for more effective exploration of the data space and the final prediction is still achieved through the deep learning model, which is designed to handle high-dimensional data well. We have compared the proposed model with random selection of unlabeled samples and more advanced attention kernels/similarity measures specifically designed for high-dimensional data with key properties, including spatial invariance. We have also carefully studied the impact the characteristic length scale of the KDE.
3.	We have conducted a detailed ablation study to clearly demonstrate the contribution from the key technical components, including vacuity, dissonance, and uncertainty anchor sample identification. We have also tested different vacuity/dissonance weighting in the proposed sampling function.
4.	We have added the batch AL results and visualized the image samples selected by ADL in the early and later stage of active learning to help better understand the role of vacuity and dissonance in data sampling.

At this point, all the major changes have been added to the supplemental materials of the paper. We will move them to the main paper for the camera-ready version if accepted. In addition, given the limited time, the results are reported for the case that AL starts with 5 classes. Our existing experiments show a similar trend when AL starts with more classes. We will add the complete results for the camera-ready version of the paper if accepted. All the code and datasets will be released after the paper is published.

---

### Decision · Program_Chairs · 2019-12-19

**Decision:**

Reject

**Comment:**

The authors propose a new perspective on active learning by borrowing concepts from subjective logic. In particular, they model uncertainty as a combination of dissonance and vacuity; two orthogonal forms of uncertainty that may invite additional labels for different reasons. The concepts introduced are not specific to deep learning but are generally applicable. Experiments on 2d data and a couple standard datasets are provided.

The derivation of the model is intuitive but it's not clear that it is "better" than any other intuitively derived model for active learning. With the field of active learning having such a long history, the field has moved towards a standard of expecting theoretical guarantees to distinguish a new method from the rest; this paper provides none. Instead anecdotal examples and small experiments are performed. Like other reviews, I am extremely skeptical about the use of KDE which is known to have essentially no inferential ability in high dimensions (such as in deep learning situations where presumably images are involved). It is hard not to feel as though deep learning is somewhat of a red herring in this paper.

I recommend the authors lean into understanding the method from a perspective beyond anecdotes and experiments if they wish for this method to gain traction.